# The Development and Validation of the Comprehensive Assessment of Acceptance and Commitment Therapy Processes (CompACT)—Malay Version

**DOI:** 10.3390/ijerph19159624

**Published:** 2022-08-05

**Authors:** Nurfarahin Musa, Nicholas Tze Ping Pang, Assis Kamu, Chong Mun Ho, Cerith Waters, Jennifer Berrett, Nima Moghaddam, Walton Wider

**Affiliations:** 1Faculty of Medicine and Health Sciences, Universiti Malaysia Sabah, Kota Kinabalu 88400, Malaysia; 2School of Psychology, Cardiff University, Cardiff CF10 3AT, UK; 3Trent Doctorate in Clinical Psychology, University of Lincoln, Lincoln LN6 7TS, UK; 4Faculty of Business and Communications, INTI International University, Nilai 71800, Malaysia

**Keywords:** CompACT, psychological flexibility, acceptance, validation, defusion

## Abstract

Objectives: psychological flexibility is a crucial construct highly correlated with psychological wellness. There is a need for a tool to measure psychological flexibility in order to accurately ascertain the effects of treatment. The existing industry standard, the Acceptance and Action Questionnaire-II (AAQ-II), has issues with conflating psychological flexibility with distress; moreover, it does not cover the hexaflexes. The 23-item CompACT was designed to surmount these limitations. Methods: the classical test theory (CTT) and Rasch measurement theory (RMT) were used to check the validity and reliability of the Malay version of the CompACT Scale. Cronbach’s α, McDonald’s Ω, and greatest lower bound were used to measure internal consistency. A Pearson’s correlation test was used to measure test–retest reliability of the Malay versus the original English version. For validity, convergent validity was established by using the Malay AAQ-7 Scale. The dimensionality of the Malay version of the CompACT Scale was explored using exploratory factor analysis. For the RMT, weighted fit statistics (infit) and outlier sensitive fit statistic (outfit) mean square (MnSq) values were used at the item level, while item and person separation reliability values and item and person separation indices were applied at the scale level. Results: the internal consistency measures, including Cronbach α and McDonald’s Ω, passed the suggested cutoff points. Convergent validity with the AAQ-II was 0.693. The quality of the Malay version of the CompACT Scale was also satisfactory, as all item and person reliability values and indices exceeded the suggested cut-off points. Conclusions: the Malay CompACT is a psychometrically sound instrument to assess psychological flexibility in both clinical and research settings.

## 1. Introduction

Psychological flexibility is defined as being in contact with the present moment, fully aware of emotions, sensations, and thoughts, welcoming them, including the undesired ones [1,2,3], and moving in a pattern of behaviour in the service of chosen values [2,4]. This is an important factor that has been found to have caused problems with several indices of psychological wellbeing. Stress, anxiety, depression, post-traumatic stress disorder, and other mental health issues can be reduced by increasing one’s psychological flexibility [5,6,7,8,9]. Interventions from the third wave of mindfulness therapies, e.g., Acceptance and Commitment Therapy (ACT), have been found to increase psychological flexibility via deployment of various psychological processes including defusion from unworkable thoughts, acceptance and being in the present moment, and identification of goals and values [10,11,12,13]. ACT suggests that illness originates from a lack of psychological flexibility, wherein people’s attempts to regulate, avoid, and escape uncomfortable thoughts and emotions act as a barrier to desired and productive behaviour [2]. ACT does not attempt to modify or diminish distressing or unpleasant internal experiences, but to restrict their effect on day-to-day life and goal attainment [14]. ACT addresses inefficient and unhelpful behaviour that avoids internal events and introduces mindfulness and acceptance-based techniques to support behaviour change and minimize psychological suffering [15,16]. The ACT method cultivates psychological adaptability by way of six interconnected positive psychological skills [17]. These skills are called “contacting the present moment”, “defusion”, “acceptance”, “self-as-context”, “values”, and “committed action”, respectively. The ACT hexaflex diagrammatically portrays these positive psychological skills. Each skill is linked to greater mental health, better health outcomes, and enhanced psychosocial outcomes across a variety of health disorders [17].

Crucially, both from research and service provision perspectives, there is a need for a tool to measure the level of psychological flexibility in individuals so that we can have quantifiable pre- and post-intervention effects, in order to accurately ascertain the effects of treatment [1]. Additionally, as asserted by Giovannetti et al., having these scales in other languages allows uniform and comparative use of this outcome measure across nations for clinical and research reasons and for the establishment of multinational trials of ACT-based therapies [18]. It is also necessary to conduct more research to validate the component structure and psychometric features of the CompACT in populations that are more culturally varied [19].

There is an established scale, called the Acceptance and Action Questionnaire-II (AAQ-II), which has been used widely in clinical practice and research settings, with reasonable reliability and validity [20,21,22,23]. However, the AAQ-II appears to conflate psychological flexibility indices with distress indices. In addition, it is seen that the AAQ-II may be somewhat unsuitable to measure the effectiveness of ACT itself. This is owing to a predominance of acceptance, fusion/defusion, and experiential avoidance-related items. This results in the AAQ-II potentially neglecting other important ACT hexaflex processes. Moreover, there are questions on whether the ACT processes can be distilled into two or only one factor, as the AAQ-II does. Thus, a longer and more multifactorial scale, the CompACT, was established in 2016 [24]. It was a 23-item instrument with a stable and theoretically coherent three-factor structure. These factors demonstrated strong internal consistency and were found to converge/diverge in theoretically congruent ways with existing measures of ACT processes, socially desirable responding, psychological distress, and general health and wellbeing.

This study, hence, aims to develop a Malay language validation of the CompACT scale using both classical test theory and Rasch analysis methods. Rasch analysis is a valuable addition to the armamentarium of tools in assessing the quality of scales, allowing us to move beyond the limitations of traditional classical test theory. This validation will greatly expand the scope of the CompACT, allowing it to be utilised in Malay language settings, and allow valuable clinical and research data to be collected, which will facilitate treatment planning and treatment progress monitoring.

## 2. Material and Methods

The convenience sampling method was used to recruit 210 undergraduate students for the survey. According to Table 1, the majority of respondents were female and single. The respondents’ average age was 24. In terms of ethnicity, more than half of the respondents were Bumiputera Sabah/Sarawak. B40, M40, and T20 Malaysia refer to the classification of household income in Malaysia. B40 represents the bottom 40% of Malaysian household income, M40 represents the middle 40%, and T20 represents the top 20%. The majority of respondents came from B40 households in terms of median monthly household income. Furthermore, the majority of respondents said they had never received treatment for a mental illness.

### 2.1. Translation Process

Communication was established with the scale’s original author, Dr. Nima Moghaddam, to obtain permission to validate the CompACT psychometric from English to Malay while adhering to WHO translation guidelines [25]. Firstly, two bilingual independent researchers were recruited for this study: one is a content expert in Acceptance and Commitment Therapy and the other is the Malay language expert. These two individuals executed forward-translation activity of the scale from English to Malay. Secondly, another two independent researchers who are also proficient in Malay and English and are experts in the content executed the backwards translation. They were not exposed to the original scale and back translated the psychometric from Malay to English. Two different versions were obtained through this method. Subsequently, the forwards and backwards translations were analysed and compared to the original version for conceptual and linguistic equivalence. Any wordings, appropriateness of format, phrasings, and cultural applicability of the translations were discussed and amended. A harmonized version was then developed. This version was then pilot tested with a group of 20 native Malay speakers who confirmed the acceptability of the harmonized translation. Any further inconsistencies or translation errors were taken into consideration and rectified. The rigorous process yielded the finalized version of the CompACT psychometric in the Malay language.

### 2.2. Measurement Scales

#### 2.2.1. CompACT Scale

Items on the CompACT were scored on a seven-point Likert scale, ranging from 0 (“strongly disagree”) to 6 (“strongly agree”). The original English version and Malay version of the CompACT Scale are shown in Table 2.

#### 2.2.2. AAQ-7 Scale

The AAQ-7 scale was administered as a measure of convergent validity to measure experiential avoidance and psychological inflexibility [26]; higher scores are correlated with psychological inflexibility.

### 2.3. Data Analysis

The classical test theory (CTT) and Rasch measurement theory (RMT) were used to check the validity and reliability of the Malay version of the CompACT Scale. For reliability, internal consistency measures using Cronbach α, McDonald’s Ω, greatest lower bound, and test–retest reliability using the Pearson correlation test (Malay version versus original English version) were used. Convergent validity was established using the Malay AAQ-7 Scale for validity. The dimensionality of the Malay version of the CompACT Scale was explored using exploratory factor analysis, which uses principal axis factor as the extraction method and oblimin as the rotation method. The analysis was conducted using IBM Statistical Package for Social Sciences (SPSS) version 27 (Developed by Norman H. Nie, Dale H. Bent, and C. Hadlai Hull, Chicago, IL, USA) and JASP Version 0.16 (University of Amsterdam, Amsterdam, The Netherlands).

For the RMT, weighted fit statistics (infit) and outlier sensitive fit statistic (outfit) mean square (MnSq) values were used at the item level, while item and person separation reliability values and item and person separation indices were applied at the scale level. MnSq values close to 1 suggest a good model–data fit [24]. The accepted range of the infit and outfit MnSq values is between 0.5 and 1.5. The recommended item and person reliability values are 0.7 or higher, while the recommended item and person separation indices are 2 or higher [27]. The Rasch analysis was conducted using jMetrik 4.1.1 (jMetrik Item Analysis, Charlottesville, VA, USA).

## 3. Results

### 3.1. Dimensionality of the Malay Version of the CompACT Scale

The Kaiser–Meyer–Olkin measure verified the sampling adequacy for the factor analysis, as the value was more than 0.5 (0.761). Bartlett’s test of sphericity (X2 (253) = 1422.075, *p* < 0.001) also confirmed that relationships existed between at least some of the items, indicating that the correlation structure was adequate for the factor analysis. The principal axis factor confirmed the dimensionality of the Malay version of the CompACT Scale, as there were three factors extracted. The eigenvalues for factor 1, factor 2, and factor 3 were 3.659, 2.881, and 1316, respectively. The three factors could explain 34.2% of the variation in the items. All the factor loadings were higher than 0.3 except item 18, as shown in Table 3.

### 3.2. Reliability and Validity of the Malay Version of the CompACT Scale

All psychometric measurements are shown in Table 4. The internal consistency measures, including Cronbach α and McDonald’s Ω, confirmed the validity and reliability of the Malay version of the CompACT Scale, as all values passed the suggested cut-off points.

### 3.3. RMT Results

The quality of the Malay version of the CompACT Scale was also satisfactory, as all item and person reliability values and indices exceeded the suggested cut-off points (refer to Table 5), except for factor 3, where the person separation reliability value was less than the cut value. The person separation indices for factor 2 and factor 3 were also less than the cut value. Low person separation indicates the possibility that the factors’ items may not be sensitive enough to distinguish between high- and low-performers. Perhaps more items are required. However, all fit statistics values, as shown in Table 6, fall within the acceptable range of the infit and outfit MnSq values, which is between 0.5 and 1.5 for the Rasch model.

## 4. Discussion

At the end of this validation process, 1 item (Item 18) was dropped due to correlations below 0.300, thus, we were left with a 22-item Malay CompACT. Using both classical test theory and Rasch analysis, we demonstrate that the 22-item Malay CompACT has suitable psychometric properties based on measures of convergent validity and multiple measures of internal consistency. In addition, despite having positively and negatively valenced items in the same scale, factor analysis reveals three factors which broadly correspond to the same domains as in the English CompACT [24]. This thus demonstrates that scales inclusive of differently valenced items can be stable and theoretically coherent.

As mapped out by Hayes et al., the three-factor structure of the English CompACT is congruent with the existing three dyadic processes [2,20], namely “openness to experience and detachment from literality” (acceptance; defusion); (2) “self-awareness and perspective taking” (present moment awareness; self as context); and (3) “motivation and activation” (values; committed action) [24]. Nevertheless, the specific items mapped to the three factors in the Malay CompACT vary slightly from the three factors in the English CompACT. For instance, two questions from “openness to experience” factor, namely: “Thoughts are just thoughts—they don’t control what I do” and “I am willing to fully experience whatever thoughts, feelings, and sensations come up for me, without trying to change or defend against them” cluster together in the third factor together with all the other original items in the third “values and committed action” factor in the English CompACT. It would appear that the elimination of two items from the “openness to experience” factor addresses both cognitive fusion and experiential avoidance. The Openness to Experience subscale of the Malay version of the CompACT appears to predominantly correspond to experiential avoidance, with less characteristics of cognitive fusion. This suggests that there are particular items that appear to correspond to different parts of the ACT hexaflex, which may suggest that the ACT hexaflex is potentially interpreted differently in different cultural contexts [28].

Of the CompACT’s subscales, it was found in the English version that openness to experience demonstrated the strongest association with the AAQ-II. The items in the “openness to experience” factor in the Malay CompACT are fewer than those in the English version; however, it still demonstrated the strongest association. Item 18 (*Even when something is important to me, I’ll rarely do it if there is a chance it will upset me*) was removed from the final scale due to its lower correlation of less than 0.300. The findings suggest that the item had slightly different connotations for East Malaysian undergraduate students. Future researchers may explore altering the item to improve its clarity and reinforce its relatedness. Malaysia is characterised by its multiethnicity, multiculturalism, and multilingualism, and its people maintain distinct cultural identities and religious beliefs [29]. As a result, further investigation needs to be conducted that goes beyond the Sabahan and East Malaysian perspectives, particularly in clinical populations that have various characteristics that correspond to them.

There are huge implications, with high possibilities of contribution on the theoretical and practical field of public health. Many studies in a Malaysian setting have already determined that a higher level of psychological flexibility is one of the key factors that contributes towards lower levels of depression, anxiety, and stress, in both cross-sectional and experimental studies. This relationship has been proven both in the general public, in specific front-liner populations, and also in a small clinical population. Hence, there is utility in establishing a more comprehensive psychological flexibility measurement instrument, given the limitations of the current AAQ-II as an overly abbreviated instrument that cannot pick up subtleties in psychological flexibility.

The limitations of this study are that it was performed primarily in a group of undergraduate students. Hence, the outcome may not be generalizable to clinical populations, and a separate validation may be needed in a separate hospital or outpatient clinic-based population. Moreover, as there is only one other extant psychological flexibility scale in practise, there were a limited number of scales on which to perform convergent validity assessments. In conclusion, the Malay version of the CompACT appears to be a reliable, valid, and quality instrument to measure psychological flexibility in a way that covers the breadth and length of the ACT hexaflex.

## Figures and Tables

**Table 1 ijerph-19-09624-t001:** Respondents’ background information (N = 210).

		Mean	Standard Deviation	N	%
Age		24	4		
Gender	Male			51	24.29%
	Female			159	75.71%
Marital status	Single			153	72.86%
	Married			12	5.71%
	In a relationship			45	21.43%
Ethnicity	Malay			41	19.52%
	Chinese			22	10.48%
	Indian			17	8.10%
	Bumiputera Sabah/Sarawak			118	56.19%
	Others			12	5.71%
Household median monthly income	<RM6275 (B40)			145	69.05%
	RM6276—RM13,148 (M40)			55	26.19%
	>RM13,149 (T20)			10	4.76%
Have you ever gotten treatment for a psychiatric illness?	No			196	93.33%
	Yes			14	6.67%

**Table 2 ijerph-19-09624-t002:** Original English and Malay versions of the CompACT Scale.

No.	Original Version	Malay Version
Item 1 *	I can identify the things that really matter to me in life and pursue them	*Saya boleh mengenal pasti perkara yang benar-benar penting dalam hidup saya dan menceburinya*
Item 2	One of my big goals is to be free from painful emotions	*Salah satu matlamat besar saya adalah untuk bebas daripada perasaan yang menyakitkan*
Item 3	I rush through meaningful activities without being really attentive to them	*Saya terburu-buru melakukan aktiviti yang bermakna tanpa benar-benar menumpukan perhatian ke atasnya*
Item 4	I try to stay busy to keep thoughts or feelings from coming	*Saya cuba untuk kekal sibuk untuk mengelak perasaan atau fikiran yang datang*
Item 5 *	I act in ways that are consistent with how I wish to live my life	*Saya bertindak secara selari dengan bagaimana saya ingin jalani hidup saya*
Item 6	I get so caught up in my thoughts that I am unable to do the things that I most want to do	*Saya asyik terperangkap dalam fikiran saya sehingga tidak berupaya untuk melakukan sesuatu perkara yang paling saya ingin lakukan*
Item 7 *	I make choices based on what is important to me, even if it is stressful	*Saya membuat pilihan hidup berdasarkan apa yang penting bagi saya, walaupun pilihan tersebut boleh membawa tekanan*
Item 8	I tell myself that I shouldn’t have certain thoughts	*Saya memberitahu diri sendiri bahawa saya tidak patut mempunyai fikiran-fikiran tertentu*
Item 9	I find it difficult to stay focused on what’s happening in the present	*Saya susah untuk menumpukan perhatian terhadap apa yang sedang berlaku pada masa kini*
Item 10 *	I behave in line with my personal values	*Saya bertindak secara selari dengan nilai-nilai peribadi saya*
Item 11	I go out of my way to avoid situations that might bring difficult thoughts, feelings, or sensations	*Saya akan berusaha untuk mengelakkan situasi yang akan membawa perasaan, fikiran atau sensasi yang menyusahkan*
Item 12	Even when doing the things that matter to me, I find myself doing them without paying attention	*Walaupun saya sedang melakukan perkara yang penting bagi saya, saya mendapati diri saya melakukannya tanpa menumpukan perhatian*
Item 13 *	I am willing to fully experience whatever thoughts, feelings, and sensations come up for me, without trying to change or defend against them	*Saya sanggup untuk mengalami sepenuhnya sebarang fikiran, perasaan, dan sensasi yang datang, tanpa mencuba untuk mengubah atau melawannya*
Item 14 *	I undertake things that are meaningful to me, even when I find it hard to do so	*Saya melalukan perkara yang bermakna bagi saya, walaupun saya mendapati perkara tersebut susah untuk dilakukan*
Item 15	I work hard to keep out upsetting feelings	*Saya bekerja keras untuk menghalang perasaan yang menyedihkan*
Item 16	I do jobs or tasks automatically, without being aware of what I’m doing	*Saya melakukan kerja atau tugasan secara automatik tanpa menyedari apa yang saya lakukan*
Item 17 *	I am able to follow my long-term plans including times when progress is slow	*Saya berupaya untuk mengikut rancangan jangka masa panjang saya, termasuk apabila kemajuannya perlahan*
Item 18	Even when something is important to me, I’ll rarely do it if there is a chance it will upset me	*Walaupun sesuatu perkara penting bagi saya, saya jarang melaksanakannya sekiranya perkara tersebut akan menyedihkan saya*
Item 19	It seems I am “running on automatic” without much awareness of what I’m doing	*Saya rasa seperti saya “berjalan secara automatik” tanpa banyak kesedaran terhadap apa yang saya lakukan*
Item 20 *	Thoughts are just thoughts—they don’t control what I do	*Fikiran cuma fikiran semata-mata. Fikiran tidak boleh mengawal apa yang saya lakukan*
Item 21 *	My values are really reflected in my behaviour	*Perlakuan saya mencerminkan nilai-nilai saya*
Item 22 *	I can take thoughts and feelings as they come, without attempting to control or avoid them	*Saya dapat menerima fikiran dan perasaan yang datang, tanpa berusaha untuk mengawal atau mengelakinya*
Item 23 *	I can keep going with something when it’s important to me	*Saya boleh teruskan sesuatu sekiranya ia penting bagi saya*

Note. * Denotes a reverse-scored item.

**Table 3 ijerph-19-09624-t003:** Factor loadings.

Item	Factor 1	Factor 2	Factor 3
Item 1 *	0.476		
Item 2		0.449	
Item 3	0.381		
Item 4		0.671	
Item 5 *	0.38		
Item 6	0.703		
Item 7 *			0.318
Item 8		0.532	
Item 9	0.717		
Item 10 *	0.503		
Item 11		0.607	
Item 12	0.577		
Item 13 *			0.673
Item 14 *			0.396
Item 15		0.499	
Item 16	0.458		
Item 17 *	0.351		
**Item 18**	0.181		
Item 19	0.552		
Item 20 *			0.505
Item 21 *	0.339		
Item 22 *			0.609
Item 23 *	0.467		

Note. * Denotes a reverse-scored item. Bold item removed from final scale due to low loading.

**Table 4 ijerph-19-09624-t004:** Psychometric properties of the Malay version of the CompACT Scale at the scale level (n = 210).

Psychometric Measure	Factor 1	Factor 2	Factor 3	Suggested Cut-Off
Internal consistency measure using Cronbach α	0.775	0.695	0.644	>0.7
Internal consistency measure using McDonald’s Ω	0.792	0.684	0.627	>0.7
Internal consistency measure using the greatest lower bound	0.895	0.761	0.716	>0.7
Convergent validity (Malay AAQ-7 Scale versus Malay version of the CompACT Scale)	0.693 **	0.348 **	−0.186 **	See Note

Note. ** The correlation is significant at the 0.01 level (two-tailed test). Correlation coefficients of <0.25 are considered as small; 0.25–0.50, moderate; 0.50–0.75, good; and >0.75, excellent.

**Table 5 ijerph-19-09624-t005:** Scale quality statistics based on the Rasch model (n = 210).

Psychometric Measure	Factor 1	Factor 2	Factor 3	Suggested Cut-Off Point
Item separation reliability value	0.9860	0.8672	0.9713	≥0.7
Item separation index	8.3870	2.5552	5.8175	≥2
Person separation reliability value	0.8082	0.7005	0.6614	≥0.7
Person separation index	2.0529	1.5293	1.3976	≥2

**Table 6 ijerph-19-09624-t006:** Item statistics of the Malay version of the CompACT Scale based on the Rasch model (n = 210).

Factor	Item	Infit MnSq	Outfit MnSq	Difficulty
Factor 1	Item 1	1.12	1.07	0.66
	Item 3	1.17	1.22	−0.34
	Item 5	Dropped	Dropped	Dropped
	Item 6	0.78	0.81	−0.17
	Item 9	0.69	0.69	−0.35
	Item 10	Dropped	Dropped	Dropped
	Item 12	0.85	0.85	−0.59
	Item 16	0.91	0.89	−0.23
	Item 17	Dropped	Dropped	Dropped
	Item 18	1.43	1.46	−0.34
	Item 19	0.80	0.78	−0.27
	Item 21	1.31	1.31	0.74
	Item 23	1.02	0.94	0.88
Factor 2	Item 2	1.14	1.13	−0.24
	Item 4	0.82	0.85	0.21
	Item 8	1.06	1.09	−0.09
	Item 11	0.98	0.96	−0.10
	Item 15	1.00	1.05	0.23
Factor 3	Item 7	1.11	1.12	0.35
	Item 13	0.83	0.82	−0.09
	Item 14	1.09	1.06	0.46
	Item 20	1.14	1.20	−0.62
	Item 22	0.86	0.87	−0.10

Note. MnSq = mean square error.

## Data Availability

The data presented in this study are available on request from the corresponding author.

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
