# Peer review of "The Development and Validation of the Comprehensive Assessment of Acceptance and Commitment Therapy Processes (CompACT)—Malay Version"

_ijerph, 2022, doi:10.3390/ijerph19159624_

Round 1

Reviewer 1 Report

The manuscript presents structured and adequate design. However, it needs adjustments in order to guarantee greater robustness to the text.

The study is specific to the area of psychology. General aspects related to public health were not expressed in writing, which can be better consumed in specific periodicals in the area. Therefore, it is recommended that the introduction and discussion be strengthened, revealing the relevance and contribution to public health from the use of the scale.

Discussion is very limited. The conceptual aspects of each analyzed domain were not explored, restricting only the psychometric properties. But what is the dialogue with the international community about the limits, possibilities and contributions of the analysis of psychological flexibility and the theoretical and practical field of public health? These aspects need to be matured.

Author Response

Dear Reviewer 1,

We are grateful for your consideration of this manuscript, and we also very much appreciate your suggestions, which have been very helpful in improving the manuscript. All the comments we received on this manuscript have been taken into account in improving the quality.

Reviewer 2 Report

This paper created a Malay version of the CompACT in order to measure psychological flexibility that covers breadth and length of the ACT hexaflex. This version was created by two sets of bilingual independent researchers where the first set focused on the forward translation (English to Malay) and the second set on the backward translation (Malay to English). After several steps of fine tuning and testing the CompACT psychometric with 20 native Malay speakers, they arrived at the final version. Afterwards they implemented the classical test theory (CTT) and Rasch Measure theory (RMT) to test the reliability of the Malay version. As a result, they concluded that the 22-item Malay CompACT has the appropriate psychometric properties to measure psychological flexibility.  

Strength:

The concept of developing different versions of the CompACT is imperative as it is crucial to have the ability to test anyone’s psychological flexibility.    

Weaknesses:

Cannot be generalized for clinical populations as this test was conducted on a group of undergraduate students. 

Author Response

Dear Reviewer 2,

We are grateful for your consideration of this manuscript, and we also very much appreciate your suggestions, which have been very helpful in improving the manuscript. All the comments we received on this manuscript have been taken into account in improving the quality.

For the weaknesses part, we are planning to run the validation on the clinical population in other article. For your information, we are the only Contextual Behaviour Science (CBS) research group in Malaysia. Our past studies and future projects can be found via this link: https://cbsresearchgroup.wordpress.com/ 

Reviewer 3 Report

Thank you for the opportunity to review your manuscript, which is a brief report of a developmental/validation study of the Malay version of the CompACT.  I notice that this is one of several translations of the original measure with versions in Chinese, German, Italian, Persian, Portuguese, Romanian, Spanish, Swedish and Tamil available. This limits the novelty of the report, but speaks to the ongoing use of the measure across multiple culture/language contexts.

I have some specific comments either because there are issues that must be addressed or for you to take into consideration.

Title

For those not immediately familiar with the CompACT, the title could be confusing because it appears to indicate a study that involved the comprehensive assessment of ACT processes rather than a measure of this name. To avoid confusion, it could be useful to adopt the same naming format as for the original version: The development and validation of the Comprehensive assessment of Acceptance and Commitment Therapy processes (CompACT) – Malay version.

Abstract

Lines 18-20. “For reliability, internal consistency measures using Cronbach α, McDonald’s Ω, greatest lower bound, and test-retest reliability using Pearson correlation test (Malay version versus original English version) were used.”  The wording here is awkward. You have “internal consistency measures using” and “test-retest reliability using” “were used”. How about this?

·         Cronbach’s α, McDonald’s Ω, and greatest lower bound were used to measure internal consistency. A Pearson’s correlation test was used to measure test-retest reliability of the Malay versus the original English version.

Lines 20-21. “For validity, convergent validity (versus Malay AAQ-7 Scale) was used.” Convergent validity is determined or established rather than used. Reword for clarity.

Line 37. In what sense do you mean that this construct is crucial? A construct is by nature conceptual and not based on empirical evidence, and you're indicating associations rather than causation, so you'll need to clarify in what sense psychological flexibility is crucial.

Material and Methods.

Line 69. A claim in the Limitations section indicates participants were predominantly undergraduate students, so perhaps more a convenience sample of students supplemented with responses from the general population?

Line 71. Repetition from line above. Also, the percentages reported throughout this p'graph repeat table content, which should be avoided.

Lines 72-74. With respect to the respondents, less than 20% identified as Malay, and ~24% were Chinese, Indian or Others. At 56%, the majority of respondents were Bumiputera Sabah/Sarawak, which brings into question if people in the target language group (i.e., Bahasa Malayu) were reached. There are numerous languages spoken in Sabah and Sarawak – how can the researchers be sure that their sample represents a majority Malay-speaking population? I wonder if attempts were made to recruit Malay speakers as an inclusion criterion and if not, why not?

Lines 75-76. What do B40, M40 and T20 represent? If no room in word count, at least insert explanation in a note beneath the table.

There is no mention of ethics approval for the study – was ethics approval sought/gained? What was the nature of participants’ consent to the use of their data?

Lines 115-118. Same comment as for the Abstract.

Results

Table 3. Re note indicating items in bold are those retained in the final scale. None of the items appear to be in bold. Also, since 23 items listed and 22 retained, why not simply bold the one item to be removed?

Lines 154-157. You refer to “all…values and indices” exceeding cut-offs, and “all scales” meeting minimum quality requirement, but three of the values in Table 5 are less than the suggested cut-off point.

Table 6. On what basis were items 5, 10 and 17 dropped? No explanation provided.

Discussion

Line 163. You haven't yet indicated why three other items were dropped, and if they were dropped just from the Rasch analysis (if so, on what basis?) or if from the measure. This latter doesn't appear to be the case based on indications of a 22-item measure.

Line 187. With respect to items being “less”, do you mean there are fewer items than in the English version?

Line 188. With respect to a ‘discovery’, the dropping of an item isn't a discovery so much as an outcomes-based decision.

Lines 191-192. This is a very broad claim that seems to suggest a similarity amongst all Asian cultures. I don't see how the citation indicated here provides sufficient evidence or grounding for such a claim, particularly as the study draws on data from a highly restricted pool: "50 Dusun first language speakers fluent in Malay from villages in rural Sabah."

Line 194. This goes against the description in the Method section indicating "individuals from the general population".

Line 203. So, you accept that there are intracultural differences as well as intercultural differences? Even amongst Asian cultures?

References

There are issues with formatting in number of the entries. I recommend the use of bibliographic citation management software such as EndNote or Zotero. 

Of 19 references, 8 are citations of publications involving at least one of the manuscript authors, and not all are the most relevant-for-purpose citations. For instance, a description of psychological flexibility [1 & 2] would be better accompanied by citation of Hayes et al.

The relevance of entry 14 isn’t clear with respect to where it’s cited in text.

Lines 82-84. “Communication was established with the scale’s original author, Dr. Nima Moghaddam, to obtain permission to validate the compACT psychometric from English to Malay while adhering to WHO translation guidelines [14,15].” Item 14 is as follows:

Pang NTP, Masiran R, Tan K-A, Kassim A. Psychological mindedness as a mediator in the relationship between dysfunctional coping styles and depressive symptoms in caregivers of children with autism spectrum disorder. Perspect Psychiatr Care. 2020;

I have uploaded a marked up copy of the manuscript indicating various places where corrections are required with respect to grammar/spelling/punctuation, etc. 

Author Response

Dear Reviewer 3,

We are grateful for your consideration of this manuscript, and we also very much appreciate your suggestions, which have been very helpful in improving the manuscript. All the comments we received on this manuscript have been taken into account in improving the quality.

Round 2

Reviewer 3 Report

Thanks for addressing my concerns. The revised manuscript addresses most of these. I've attached a marked-up copy of the revised manuscript with highlights indicating where I've made some further comments or queries. These are minor points. 

Author Response

(The authors gave the same response as above.)
